# The Impact of Long-Term Physical Salt Attack and Multicycle Temperature Gradient on the Mechanical Properties of Spun Concrete

**DOI:** 10.3390/ma14174811

**Published:** 2021-08-25

**Authors:** Romualdas Kliukas, Arūnas Jaras, Ona Lukoševičienė

**Affiliations:** Department of Applied Mechanics, Faculty of Civil Engineering, Vilnius Gediminas Technical University, 10223 Vilnius, Lithuania; arunas.jaras@vilniustech.lt

**Keywords:** spun concrete, durability, PSA, temperature, cyclic wetting-drying, chemical admixtures

## Abstract

The article is focused on spun concrete made with different chemical admixtures under long-term exposure to aggressive salt-saturated ground water and a cyclic temperature gradient. Over a long-term experimental investigation, 64 prismatic spun concrete specimens were subjected to multicycle (75–120) processing under combined aggressive ambient conditions. Prismatic specimens were soaked in water or saline and dried at a temperature of 45–50 °C. The long-term multi-cycle effect of the temperature gradient and physical salt attack on the compressive strength, Young’s modulus and durability of concrete was found to be negative. Chemical admixtures, though, improved the structure of spun concrete, thus having a significant positive effect on its physical-mechanical properties and durability.

## 1. Introduction

Corrosion of the protective layer of concrete reinforcement and the subsequent corrosion of the reinforcement are the main reasons that reinforced concrete structures deteriorate before their time. Metal corrosion products peel off the protective layer, the corrosion of the metal reduces the cross-section of longitudinal and transverse reinforcement, and the transverse reinforcement breaks, resulting in irreversible decay.

The scientific literature deals with a number of factors related to the corrosion of concrete: residual water–cement ratio (W/C)_res_, hardening conditions, compaction degree, alternating wetting and drying conditions, freezing and thawing, bacterial corrosion and attacks by various chemicals (e.g., chlorides and sulphates). However, the corrosion of concrete by physical salt attack from saline groundwater is one of the most dangerous and aggressive factors and leads to significant financial losses.

Deterioration from physical salt attack (PSA) occurs when crystallized groundwater salts (sodium sulphate, sodium carbonate and sodium chloride) accumulate in concrete pores and build up pressure. Such salt weathering is most commonly observed in arid climates such as in North Africa, the Middle East, a few American states and Australia [1].

The cyclic wetting and drying of saline in hardened cement paste appears to cause the most destruction because it allows salts to migrate to the evaporating surface through capillary action. Subject to the structure of the paste, salt crystallization can occur directly on its surface or within a certain layer. In the first case, the concentration of calcium hydroxide in the adjacent layer of hardened cement paste gradually decreases since its solubility increases in the presence of salts, thereby causing type 1 corrosion.

If the size of the pores is large enough (≥0.2 μm), water evaporation occurs on the surface of the product and in the pores of the surface layer of the concrete. Next, salt crystallizes in the pores of the hardened cement paste, leading to its deterioration (corrosion type 3).

Although the mechanism of concrete fracturing is not completely clear, there are several hypotheses about the causes of the deterioration of hardened cement paste under the impact of saline in the presence of surface evaporation [2].

One of the first hypotheses about the salt crystallization pressure within the pores of hardened cement paste was that it was due to growing tensile stresses. The reliability of this hypothesis is questionable on the grounds that the crystallization pressure is insignificant and cannot be the only reason for the cement paste deterioration.

A second hypothesis suggests that deterioration caused by potassium salts occurs due to an increase in the volume of neoplasms. As a result, along with the accumulation of salts in pores, internal stresses arise that cause crack formations, which contribute to the penetration of an aggressive medium deep into the concrete. Deterioration may also occur because of the wedging action of water when concrete is saturated with saline, which generates alternating stresses under cyclic wetting and drying, osmotic forces and differences in the coefficients of the thermal expansion of hardened cement paste and salts.

According to a third hypothesis, the main reason for the deterioration of hardened cement paste subjected to cyclic wetting with saline and drying is an increase in the volume of salt crystals at the conversion phase from less to more hydrated. Drying concrete at a temperature above the point of phase conversion of salts followed by wetting it at a lower temperature, accompanied by the formation of expanding crystalline hydrates, is accepted as the most destructive state.

None of the above hypotheses provides an exhaustive explanation for the deterioration process of hardened cement paste under the aggressive action of saline characterized by the complexity of the phenomena in this particular case. Apparently, the corrosion mechanism is determined by a combination of all the abovementioned factors. The natural operating conditions of structures frequently involves several unfavourable factors such as aggressive saline groundwater, temperature fluctuations during the day, and dry winds causing the capillary migration of saline groundwater into the concrete of the structure.

The scientific literature provides a number of studies on factors having an impact on concrete durability, including physical salt attack (PSA) [3,4,5,6,7,8,9,10,11,12,13], cyclic wetting and drying (CWD) [14,15,16], the water–cement ratio of the initial concrete mix [14,17,18,19], chemical admixtures [20].

The clear majority of studies deals with durability and reliability of vibrated concrete [21,22,23,24,25,26,27,28,29,30,31,32], and significantly less on the durability of spun concrete [33,34,35,36,37] because the latter is much more expensive, time consuming and complex.

The impact of PSA and CWD on the physical–mechanical properties of vibrated concrete is considered in the research papers that state that exposure to saline initially improves the structure of concrete [14,37], while the damaging effects of pressure from salt crystals growing in concrete pores are noticeable only after long-term exposure to CWD.

Experimental results presented by Liu et al. [14] showed that following 80 CWD cycles of a sulphate attack, the compressive strength of ordinary concrete increased by 9.1 and 6.1% (subject to the water–cement ratio), while in the succeeding 120 CWD cycles, compressive strength decreased by 15.6 and 11.3%, respectively. Concrete soaking accelerated the corrosion caused by cyclic wetting and drying [14,37].

The operating temperature of a concrete structure and its gradient during the day is another very significant factor that determines its durability and is closely related to the abovementioned one. The surface temperature of a structure often varies from +60 °C during the day to −20 °C at night time (e.g., in the deserts of Asia, North Africa or Australia) and leads to elongation–shortening (swelling–shrinkage) strains leading to concrete destruction. The same swelling–shrinkage strains of hardened cement paste due to the wind-induced capillary filtration of groundwater are caused by wetting–drying in the contact zone with the ground.

A detailed analysis of climate changes having an impact on the deterioration of concrete infrastructure is presented in [38]. Despite the fact that the analysis involved an example from Australia, inherent problems are typical of all drylands.

The researchers investigating the impact of an operating temperature on the properties of vibrated concrete noted that the long-term effects of the temperature gradient on the mechanical properties of concrete was negative and subject to the composition of the concrete mix (water–cement ratio), the magnitude of the temperature gradient and the frequency of variations, groundwater aggressiveness (chemical composition), capillary soil–water migration in concrete, the humidity of the environment, and wind intensity [39,40,41,42,43,44].

In a review of the literature, D.J. Naus found that concrete showed a slight loss of strength when the steady-state temperature rose from 20 to 200 °C. In the case of seasonal and daily periodic variations in temperature, the situation was quite different [39]. Cyclical variations and changes in CWD-induced strain in concrete accelerated the occurrence and development of damage, degraded its mechanical properties and led to destruction. D. Chen et al. noted that when the temperature gradient ranged between 1.1 and 32.2 °C (reflecting real climatic conditions) the loss rate of the Young’s modulus was lower by over 12% [40]. With reference to the results presented by V. Korovyakov et al., one of the ways of optimizing the cracking resistance of concrete in a hot (up to 50 °C) and arid climate is to increase the amount of cementitious components when the W/C ratio does not exceed 0.34 [44]. An investigation of recycled aggregate concretes exposed to high (200, 400 °C) temperatures can be found in [45].

The adverse atmospheric factors (saline groundwater and temperature variations during the day) frequently affected the reinforced spun concrete supporting poles of overhead power transmission lines (Figure 1).

The previous studies conducted by the authors demonstrated that the compressive strength and Young’s modulus of spun concrete decreased by 20–40% following 75 cycles of wetting in water and drying in the air at 100 °C, and the mechanical properties decreased to a similar extent. Actually, this was significantly influenced by a large temperature gradient (Δt ≈ 80 °C) formed during the wetting–drying cycle [37].

Using concrete admixtures to produce denser concrete is one method of increasing the service life of reinforced concrete structures under the aggressive ambient conditions mentioned above. The selection of appropriate admixtures depends on the granulometric and chemical composition of coarse aggregates, cement and sand in the area, the method of compaction and hardening of reinforced concrete elements, and the chemical admixtures produced mainly from industrial waste in the region [46,47,48]. Therefore, each case requires an experimental study albeit expensive and labour intensive but with a high economic impact.

This article analyzes the impact of long-term physical salt attacks (PSAs) and an exclusively multicyclical temperature gradient on the mechanical properties of spun concrete modified by chemical admixtures. The research to be continued also covers other unfavourable impacts on spun concrete.

## 2. Materials and Technique of Investigation

### 2.1. Materials and Concrete Mixture Design

#### 2.1.1. Cement

Portland slag cement having a compressive strength of 40 MPa was used. Its main physical and mechanical properties are presented in Table 1 and discussed in more detail in [37].

#### 2.1.2. Aggregates

To prepare the concrete mix, sand consisting of particle sizes from 0.14 to 1.25 mm was used as a fine aggregate, and granite rubble of 2.50 to 25.00 mm was employed as a coarse aggregate. The maximum content of particles from 20 to 25 mm in diameter made was 10% by weight. The maximum particle size of the coarse aggregate conformed to technological and design requirements of not exceeding 1/3 of the wall thickness of thin-walled specimens. The physical and mechanical properties of the aggregates are given in Table 2.

#### 2.1.3. Water

Potable water from the water supply system was used to produce the concrete mix.

#### 2.1.4. Chemical Admixtures—Superplasticizers

Three types of superplasticizers were used to investigate the impact of chemical admixtures on the physical and mechanical properties of spun concrete (Table 3).

The effective amount of the admixtures was based on previous research performed by the authors [33]. All admixtures were added to the concrete mixes as a water solution.

#### 2.1.5. The Composition of the Concrete Mix

For durability studies on spun concrete, specimens were obtained from heavy concrete having a compressive strength of 60–70 MPa. The sand, crushed stone, cement and water components in concrete mixtures were 400, 1280, 565 and 164–209 kg/m^3^, respectively. Due to the centrifugation process, part of water and cement admixtures were washed from concrete mixtures. Therefore, the initial water–cement ratio varied from 0.29 to 0.37 and decreased to the residual water–cement ratio of 0.28–0.30. The compositions of the concrete mixes, including chemical admixtures, were selected to make the consistency of the concrete mix similar to that of the concrete mix prepared without admixtures. This was achieved by varying the water–cement ratio. The contents of the chemical admixtures and the initial water–cement ratio (W/C)_in_ in the concrete mix are given in Table 4.

The composition of concrete mixtures prepared without chemical admixtures corresponded to those used to manufacture spun poles for overhead power lines and provide a compressive strength of spun concrete of 60 MPa.

### 2.2. Manufacturing Specimens and Investigation Technique

Manufacturing specimens to test the effect of salts and temperature on the resistance of spun concrete covered 2 stages. This study used 42 thin-walled circular cross-section specimens, the main geometrical parameters of which were a height of 550 mm, outer diameter of the annular cross section of 560 mm, and wall thickness from 75 to 90 mm. In this case, metal moulds with an inner diameter of 560 mm and a length of 23,100 mm for producing the spun concrete supporting poles were employed. To mould the prototypes to a predetermined height of 550 mm, special metal diaphragms were used.

The metal mould contained 4 types of concrete mixes, 11 specimens of which were obtained from the admixture-free concrete mix using the Dofen superplasticizer. Ten specimens were moulded from the concrete mix containing superplasticizers C-3 and ACF-3M.

The specimens were produced by applying a roller-type centrifuge to manufacture the supporting poles of overhead power transmission lines. To form the specimens, the following centrifugal consolidation mode was applied:rotating for 3 min until 50–80 rev/min was reached,rotating for 1 min until 150 rev/min was reached,rotating for 1 min until 200 rev/min was reached,rotating for 1 min until 300 rev/min was reached,compaction for 15 min when 420–442 rev/min is reached.

The total time for the concrete compaction process employing centrifugation was 21 min. The metal mould filled with spun concrete was heat-treated in keeping with the following mode:2 h at 20 °C;2 h rising to 75 °C;3 h at +75 ± 5 °C;2 h at a falling temperature.

The heat treatment of spun concrete took place in an induction chamber. Heating the concrete, made of reinforced concrete products, was carried out by heating a metal mould and reinforcing it in an electromagnetic field.

Part of the circular cross-sectional specimens (Figure 2a) was cut into prisms (Figure 2b) with one side equal to ~200 mm and the other corresponding to the thickness of the circular specimen (75–90 mm). Each element of the circular cross-section was divided into 8 prisms (Figure 2b): 2 were stored at normal temperature and humidity conditions for further study, and 3 prisms were soaked for 75 (or 120) cycles in water and dried at 45–50 °C in the air. The other 3 prisms were soaked for the same 75 (or 120) cycles in saline and air-dried at temperatures varying from 45 to 50 °C to ensure the reliability and comparability of the experiment results.

Investigation of the resistance of spun concrete to different types of severe ambient conditions was carried out in two stages following 75 or 120 cycles of CWD.

Table 5 shows data from 64 prisms from 8 cut elements with an annular cross-section. Control prisms were cut from a specimen of the same annular cross-section, stored under natural laboratory conditions, and tested at the same time as the prisms soaked in water or saline.

The following modes of aggressive ambient conditions were applied to the abovementioned technique for testing spun concrete specimens:

1. Alternate drying of specimens under ambient air conditions for 11 h at a 45–50 °C followed by ambient cooling at 18–22 °C, and by wetting in water for 12 h at 20–25 °C.

2. Alternate drying of specimens under ambient air conditions for 11 h at 45–50 °C followed by ambient cooling at 18–22 °C, and by wetting in saline for 12 h at 20–25 °C.

To completely eliminate additional hydration of cement within the process of wetting the hardened cement paste, integrated (combined) tests on spun concrete were carried out for more than one year.

The composition of saline for creating an aggressive salt solution environment under laboratory conditions was chosen considering the results of the chemical analysis of ground water in the region of servicing reinforced spun concrete members—supporting poles of overhead power transmission lines (Table 6).

It should be noted that the severe ambient conditions (salt-saturated ground water) were in contact with only the outer side of the supporting pole. Therefore, to fully reflect the one-sided action of saline under laboratory conditions before testing both ends of the prismatic specimens, three lateral surfaces (along the cutting) were waterproofed with 5 coats of polyvinyl acetate (PVA) glue. Next, the specimens were submerged in a specially equipped container filled with saline or water (subject to the mode of aggressive ambient conditions) for up to 24 h.

## 3. The Assessment of the Mechanical Properties of the Specimens

The resistance of spun concrete to aggressive ambient conditions was estimated by the concrete resistance coefficients *α* and *β* [37]: the estimated compressive strength $f_{c}$, and the initial Young’s modulus *E*, respectively. These dimensionless coefficients characterized the degree to which the initial mechanical properties of spun concrete were preserved. The degree of preservation (resistance coefficient) was the ratio of the indicators for the mechanical properties of the exposed concrete to those of the concrete that was not exposed. The study determined the experimental values the coefficients to the impact of the various aggressive ambient conditions.

Considering that manufacturing the spun concrete specimens involved the use of different chemical admixtures, and that durability testing disclosed the varying effects of the ambient conditions, conditional three-digit indices were adopted to indicate the mechanical properties. Each index shows the type of the chemical admixture used and the temperature and wetting process of the basic specimens.

The first digit of an index denotes the grade of the chemical admixture:

0. Admixture-free concrete mix.

1. Concrete mix containing acetone-formaldehyde resin ACF-3M;

2. Concrete mix containing superplasticizer ‘Dofen’;

3. Concrete mix containing superplasticizer C-3.

The second digit denotes the temperature of environmental exposure:

0. Specimens are stored at an air temperature of 18–22 °C;

1. Specimens are dried at an air temperature of 45–50 °C.

The third number denotes the ambient conditions under the wetting mode of the main specimens at 20–25 °C:

0. Normal temperature and humidity.

1. Water.

2. Saline.

The resistance of spun concrete to temperature variations was estimated with reference to the coefficients:(1)$$\alpha_{1}=f_{c,i,1,1}/f_{c,i,0,0}$$
(2)$$\beta_{1}=E_{c,i,1,1}/E_{c,i,0,0}$$
where $f_{c,i,1,1}$ and $E_{c,i,1,1}\;$ are, respectively, the prismatic compressive strength and the Young’s modulus of spun concrete subjected to alternate wetting in water at 20–25 °C and drying in the air at 45–50 °C; $f_{c,i,0,0}$*,* and $\;E_{c,i,0,0}$ are indicators for the relevant mechanical properties under normal temperature and humidity.

The assessment of the resistance to the integrated (simultaneous) impact of several types of the aggressive ambient conditions plays an important role in analyzing the durability of concrete. The tests were aimed at investigating resistance to the integrated aggressive condition of saline and temperature variations. The durability was estimated by employing the coefficients:(3)$$\alpha_{2}=f_{c,i,1,2}/f_{c,i,0,0}$$
(4)$$\beta_{2}=E_{c,i,1,2}/E_{c,i,0,0}$$
where $f_{c,i,1,2}\;$ and $E_{c,i,1,2}\;$ are, respectively, the prismatic compressive strength and the initial Young’s modulus subjected to alternate wetting in saline at 20–25 °C and drying in the air at 45–50 °C.

Resistance to the corrosive effects of salts is one of the main determinants of durability under dry and hot climatic conditions. In deserts, the underground parts of reinforced concrete and concrete structures used as supports are subjected to concrete corrosion class 3, which combines all interactions with the environment. These processes are related to the formation and accumulation of poorly soluble salts. An increase in the volume of salts during conversion to the solid phase (i.e., the stage of crystal formation) causes internal stresses and destructive phenomena. Similar occurrences may take place as a result of the crystallization of reaction products and salts in the pores of hardened cement paste coming from the outside in the form of a solution. The rate at which the processes proceed largely depends on temperature. Thus, in the hot climate of a desert, corrosion class 3 progresses much faster than at low temperatures under other harsh climate conditions.

Unfortunately, current standards do not provide a method to determine the resistance of concrete to corrosion. In this case, the well-known technique introduced by Skramtaev [49] appears to be the most common method for an accelerated determination involving cycles of saturation in an aggressive solution of salts followed by drying. The previous tests performed by the authors [36], as well as preliminary short-term studies on the durability of spun concrete [37], demonstrated that the Skramtaev method should focus primarily on the temperature mode of drying because ignoring this factor may lead to errors caused by the negative impact of the temperature gradient on the physical and mechanical properties of spun concrete.

Thus, the authors of the present study agreed to exclude temperature when identifying the impact of salt crystallization on the mechanical properties. The achieve this goal, improvements were made to the testing methodology and technique for assessing corrosion resistance. In contrast to the resistance tests described by the authors in [37], the drying temperature was reduced from 100–105 °C to 45–50 °C, thus increasing the duration. The experimental data was assessed using the resistance coefficients and determined with reference to the following formulas
(5)$$\alpha_{3}=\alpha_{2}/\alpha_{1}=f_{c,i,1,2}/f_{c,i,1,1}$$
(6)$$\beta_{3}=\beta_{2}/\beta_{1}=E_{c,i,1,2}/E_{c,i,1,1}$$

To calculate these coefficients, the compressive strength or the initial Young’s modulus of spun concrete was prepared using the same admixture of concrete mix and the results of soaking in an equal number of cycles in saline and water were compared. After wetting, the specimens were air-dried at the same temperature.

## 4. Experimental Results and Analysis

### 4.1. Experimental Results

The main parameters of the cross-sections and the results of short-term axial compression tests on control (not subjected) prismatic specimens and those subjected to different aggressive environments are given in Table 7, Table 8 and Table 9.

The results of the previous study used to compare the results of the experimental study were published in the [37]. The experiments involved similar specimens soaked in water or saline for 25, 50 and 75 cycles and air-dried at a temperature of 100–105 °C [37].

### 4.2. The Resistance of Spun Concrete to Temperature Variations

Testing results following 75 and 120 cycles of alternate drying of concrete in the air and wetting in water show that the presence of temperature variations, although very small (Δt ≈ 25°) but lasting a long time (with a large number of CWD cycles), has a negative effect on the mechanical properties and thus durability. This is shown by the experimental values of the resistance coefficients α_1_ and β_1_ as shown in Table 10 and Figure 3.

The analysis of the results demonstrated that the chemical admixtures had a significant positive effect on the resistance of spun concrete to temperature variations. Figure 3 and Table 10 show that following 75 cycles of CWD, the mean value of the resistance coefficient *α*_1_ made with admixtures Dofen, C-3 and ACF-3M fluctuated within a range of 0.83–0.90, and coefficient *β*_1_ varied within a range of 0.91–0.99. As for traditional spun concrete, these coefficients were 0.85 and 0.80 respectively. Overall, a small temperature gradient Δt ≈ 25 °C following 75 wetting and air-dried cycles slightly reduced the strain index of concrete containing admixtures [34]. Although the Young’s modulus of admixture-free spun concrete decreased by approximately 20%, the compressive strength of that containing admixtures decreased from 10 to 15%.

The previous experiments conducted by the authors of [37] and displayed in Figure 3 showed that dramatic variations in temperature (wetting at a temperature of 20–25 °C and drying in the air at 100–105 °C), following 75 cycles of CWD of spun concrete containing superplasticizers Dofen and C-3 significantly reduced the initial water–cement ratio in the concrete mix, while compressive strength decreased by around 15% and the Young’s modulus dropped by approximately 25%. Meanwhile, the compressive strength and Young’s modulus of the control specimens decreased by 32 and 44%, respectively.

Thus, a cyclic drop in temperature from 100–105 °C to 20–25 °C had a significantly negative effect on the mechanical properties and durability of the traditional spun concrete (control), which explains the “jump” in the experimental data diagram in Figure 3.

Figure 3 and Table 10 show that following 120 cycles of CWD, the average value of the resistance coefficients *α*_1_ produced with admixtures Dofen and C-3 varied within the range of 0.75–0.78, while the resistance coefficient of the traditional spun concrete and the one containing the ACF-3M admixture was 0.64–0.69. A similar situation was observed for coefficient *β*_1_: 0.78–0.81 for concrete with admixtures Dofen and C-3 and 0.61–0.68 for admixture-free concrete or containing the ACF-3M admixture, respectively.

From the preceding, it can be concluded that a drop in the temperature gradient seen during the experiments failed to completely exclude the impact of temperature on the mechanical properties.

Alternating drying in the air and wetting in water showed that the magnitude of a negative impact on the mechanical properties was subject to the number of CWD cycles and the gradient of temperature variations throughout the wetting–drying processes.

The nature of changes to the indicators of durability depending on the number of CWD cycles suggested that the microdefects identified in the concrete structure and caused by the presence of temperature differences in the environment prevented the different types of concrete from having a denser structure [37].

After 75 cycles of CWD, concrete containing admixtures C-3 and Dofen tended be suppressed, whereas the admixture-free concrete and that produced with ACF-3M still developed intensively. At some point between 75 and 120 cycles of CWD, the above-mentioned negative impact of the temperature gradient started to manifest much more rapidly and the decrease in the resistance coefficients $\alpha_{1}$ and $\beta_{1}\;$ were more drastic. It should be noted that these tests exhibited a reduced positive effect for concrete wetting on the strength and strain properties expressed in the supplementary l hydration of hardened cement paste due to the received moisture surplus required. The testing of 75 and 120 cycles took place under the concrete age of 1.5–2 years old. Previous research showed that spun concrete of this age was stored under normal temperature-wetting conditions and only gained an annual strength of 5–7% [50].

The reason that the destruction of concrete with admixture ACF-3M developed similarly to that of admixture-free concrete can be explained by considering the composition of the concrete mix [37]. The initial water–cement ratios were ≈0.37, which meant that more microcapillaries formed in the concrete of the manufactured products along the centrifugation process and the density of concrete decreased compared to that containing admixtures C-3 or Dofen. Therefore, swelling–shrinkage strains observed within the processes of wetting–drying concrete had a greater negative impact on the stress and strain properties of lower-density concrete, thus decreasing resistance to tensile strains.

### 4.3. The Resistance of Spun Concrete to the Integrated Impact of Temperature Variations and Saline

To assess the resistance of spun concrete to the integrated effects of temperature variations and saline, data on the effect of chemical admixtures on the mechanical properties of spun concrete were obtained and are presented in Table 11 and Figure 4.

In the presence of the integrated effect of saline and temperature variations (albeit small), a negative effect of the latter on the mechanical properties was observed: the formation of microcracks along the entire structure and smoothed by the positive impact on the stress and strain properties by newly formed salt crystals in the microcracks and pores The microcracks appeared between the coarse aggregate, which accumulated on the outer surface of the product during centrifugation and on the layer of cement paste due to the effect of the cyclic temperature difference, as a result of which the saline solution penetrated. It crystallized, accumulated and caused internal pressure, thus increasing the crack and peeling of the coarse aggregate from the hardened cement paste.

The integrated effect of the above-mentioned factors on the mechanical properties was estimated by resistance coefficients *α*_2_ and *β*_2_ according to (3) and (4) respectively.

Figure 4 shows that following 75 cycles of CWD, coefficient *α*_2_ was greater than *α*_1_. Following 75 cycles of CWD, coefficient *β*_2_ was also higher than *β*_1_. In addition, coefficient *β*_2_ was greater than 1 for all types of spun concrete tested. Following 75 cycles of CWD, the compressive strength of the concrete containing the chemical admixtures decreased to 10%; meanwhile, the compressive strength of admixture-free spun concrete was 10% higher than that of the control specimens.

This confirmed the conclusion that saline––the crystallization of new formations in the pores and micro-cracks of concrete––increased the strength of spun concrete up to a certain number of CWD cycles. The same situation was observed discussing the initial Young’s modulus of spun concrete exposed to saline and temperature variations. In the case of the combined effect of the temperature gradient and saline, at some point between 75 and 120 cycles (or possibly earlier), compressive strength and the initial Young’s modulus showed a greater negative effect from the temperature gradient rather than a positive effect from salt crystallization in concrete pores. The results below confirm this statement.

As indicated by the previous experimental studies [37], Figure 4 shows that the integrated cyclic impact of a high temperature gradient of Δt = 80 °C and saline reduced the compressive strength of spun concrete modified with chemical admixtures C-3 and Dofen and decreased the Young’s modulus by around 22%, whereas the admixture-free concrete estimated a drop of approximately 35–40%.

### 4.4. Resistance of Spun Concrete to Corrosion

To assess the resistance of spun concrete to salt corrosion the coefficients were been determined considering the results of the tests performed at a reduced drying temperature. The values were calculated by applying formulas (5) and (6) and are given in Table 12 and Figure 5.

Figure 5 shows that saline had a positive effect on the strain and stress properties of spun concrete in the run of up to 75 cycles of CWD. The values of corrosion resistance coefficients were subject to the type of the chemical admixture employed. The coefficients of admixture-free concrete were equal to *α*_3_ = 1.28, *β*_3_ = 1.34, while those of concrete containing C-3 or Dofen admixtures were 1.08 and 1.07–1.10 respectively.

The increase in these indicators under the impact of showed that the use of admixtures made it possible to reduce the initial water–cement ratio of the concrete mix, which resulted in denser and more homogeneous structure (Table 4).

A spun concrete product has an outer surface that is dense and has low-porosity, which is the opposite of its inner surface. Centrifugation, a specific method for compacting a concrete mix, pushes larger particles to the outer wall of the formwork while the smaller ones face the cylindrical hole at the centre of the cross-section. In addition, centrifugation expels excess water from the concrete mix towards the inner surface. As a result, microcapillary voids are formed in the cross-section and directed from the outer surface towards the centre, which further reduces the density of the inner layer.

Salt crystallization in the pores takes place under the influence of saline, as a result of which the concrete acquires a more homogeneous structure, particularly in the inner, more porous surface. For more heterogeneous and porous concrete, salt crystallization has positive effect of on the stress and strain properties up to a certain number of cycles. The data provided in Figure 5 and Table 13 show that an increase in the compressive strength and Young’s modulus of concrete containing admixtures was caused by salt crystallization and found to be higher than that in admixture-free concrete. It should be noted that a positive effect of saline was the additional hydration of hardened cement paste due to the added moisture necessary for this process (not as high as provided above).

Temperature drops and alternate swelling–shrinkage under cyclic wetting–drying have a negative effect on the mechanical properties of concrete and may cause micro-cracks, the magnitude of which mainly depends on the temperature gradient at CWD.

However, Figure 5 shows that following 75 cycles of CWD having relatively small fluctuations in temperature, salt crystallization still had a positive effect on the stress and strain. It also shows that the resistance of spun concrete to corrosion depends slightly on the type of the chemical admixture used, which makes it difficult to identify the most effective one. It should be noted that the C-3 admixture conformed to all indications of resistance and was found to be the most effective in our case. CWD in saline at a normal drying temperature has a positive effect on the mechanical properties of spun concrete up to 75 cycles.

Under a large number of CWD cycles, the process of micro crack formation in concrete actually stops, and a positive effect of corrosion class 3 on the stress and strain properties of concrete (removing defects) turns into the negative one since an adverse effect of the pressure of growing salt crystals on the walls of pores and micro cracks in concrete is observed.

Figure 5 show that following 120 cycles of CWD, the effect of salts on concrete strength compared to the initial strength of control specimens, becomes negative and reduces compressive strength by 6–13% for all investigated types of concrete. A positive effect of salts on Young’s modulus is still quite significant for admixture-free concrete. However, for concrete containing admixtures this effect is negative compared to the Young’s modulus of control specimens.

Thus, it is assumed that a further decrease in the indicators describing the mechanical properties of concrete and the intensity of concrete destruction will be subject to the concentration of saline (groundwater), the value of the temperature gradient and the number of CWD cycles.

## 5. Conclusions

Experimental methods have involved revealing a long-term negative cyclic effect of the temperature gradient on compressive strength, Young’s modulus and durability of spun concrete. The decrease in compressive strength and Young’s modulus depended on the temperature gradient and the duration of the cyclic effect, which, along with the induced swelling–shrinkage (elongation–shortening) strains may cause damage to the concrete structure.

Following 120 cycles of concrete wetting in water and drying in the air at a temperature of approximately 25 °C, the temperature gradient on average reduced the compressive strength of admixture-free spun concrete by 35% and the Young’s modulus by 40%.

The compressive strength of spun concrete containing admixtures—superplasticizers C-3 and Dofen and decreasing the initial water–cement ratio—was reduced to 25% and the Young’s modulus to around 20%. Meanwhile, the spun concrete containing the admixture ACF-3M only slightly reduced the initial water–cement ratio of the concrete mix, which led to a drop in compressive strength and the Young’s modulus of approximately 30%.

The impact of salts dissolved in water was found to produce a dual effect on the compressive strength and strains. Cyclic wetting and drying in saline improved the physical–mechanical properties of concrete at the initial stage of the process, which is the reason that, over time, salt crystallization occurred in the pores and microcracks of concrete that had a different prehistory of development, including cracks that originated from cyclic variations in temperature. The long-term integrated effect of temperature variations and saline on concrete showed that a positive effect of corrosion class 3 on the stress and strain properties turns negative because of a more pronounced effect of the pressure of growing salt crystals on the wall of the pores and microcracks in concrete. Following 120 cycles, the compressive strength of spun concrete containing admixture ACF-3M and that of admixture-free spun concrete under the impact of temperature variations and CWD decreased by approximately 40% and that of the Young’s modulus by around 25%. The compressive strength and Young’s modulus of concrete with admixtures C-3 and Dofen were reduced by 30 and 25% respectively.

The resistance of spun concrete to corrosion was found to be mostly subject to the parameters for the concrete structure, i.e., the volume of pores and microcracks in concrete. Chemical admixtures that improved the structure of spun concrete had a significant, positive effect on the mechanical properties and hence durability.

## Figures and Tables

**Figure 1 materials-14-04811-f001:**
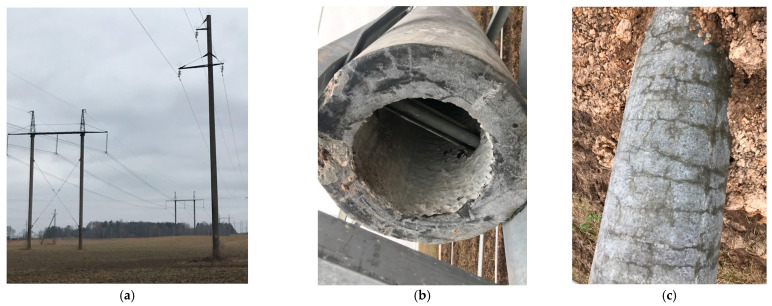
Reinforced spun concrete supporting poles of the overhead power transmission lines (**a**); cross section (**b**); lateral cracks (**c**).

**Figure 2 materials-14-04811-f002:**
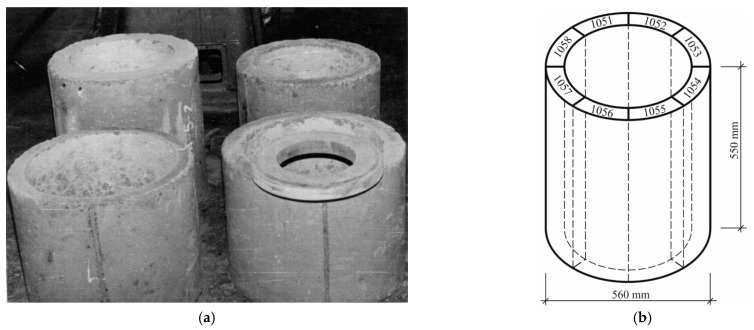
The elements of the circular cross-section (**a**) scheme for cutting element No 105 made of the concrete mix containing admixture ‘Dofen’ into prisms (**b**): prisms 1051, 1053, 1056 soaked in saline; prisms 1052, 1055, 1057 soaked in water; prisms 1054, 1058 kept under natural conditions.

**Figure 3 materials-14-04811-f003:**
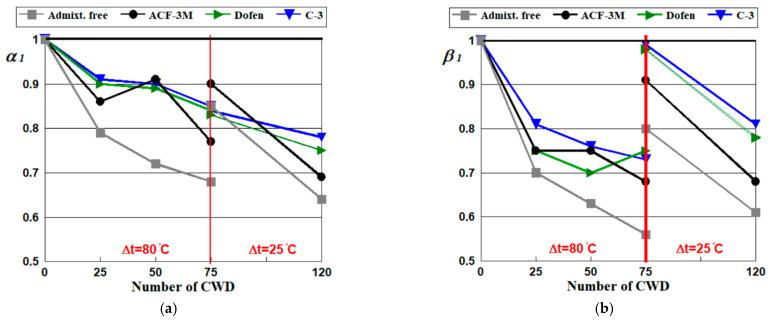
Diagrams of the relationships of the resistance coefficients of spun concrete *α*_1_ (by Equation (1)) (**a**) and *β*_1_ (by Equation (2)) (**b**) subject to the number of wetting cycles in water, drying in the air and the temperature gradient considering the type of the chemical admixture used.

**Figure 4 materials-14-04811-f004:**
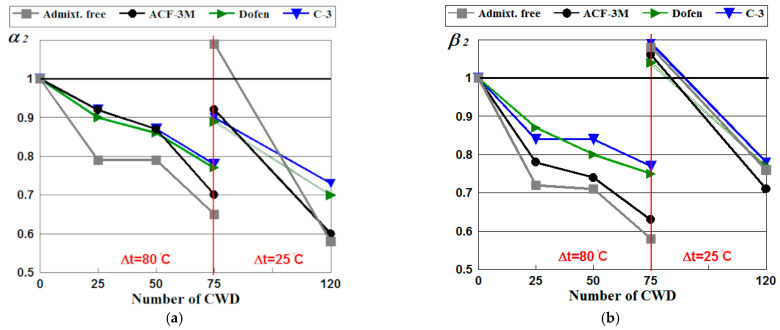
Diagrams of the relationships of the resistance coefficients of spun concrete *α*_2_ by Equation (3) (**a**) and *β*_2_ by Equation to (4) (**b**) subject to the number of wetting cycles in saline, drying in the air and the temperature gradient considering the type of the chemical admixtures used.

**Figure 5 materials-14-04811-f005:**
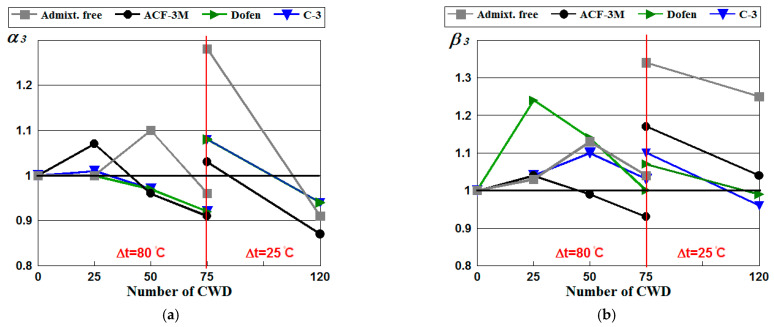
The diagrams of the relationships of the resistance coefficients of spun concrete to corrosion *α*_3_ by Equation (5) (**a**) and *β*_3_ by Equation (6) (**b**) subject to the number of wetting cycles in saline, drying in the air and the temperature gradient considering the type of the chemical admixtures used.

**Table 1 materials-14-04811-t001:** The main physical–mechanical properties of Portland slag cement.

Parameter	Unit	Value
Normal consistence of cement grout	%	26.2
Initial setting time	hour	3.25
Final setting time	hour	4.75
Griding fineness	%	85.0
Volumetric density	kN/m^3^	26.5

**Table 2 materials-14-04811-t002:** The physical-mechanical properties of fine and coarse aggregates.

Properties	Units	Coarse Aggregate	Fine Aggregate
Volumetric density	kN/m^3^	15.10	13.50
Volumetric compacted density	kN/m^3^	17.00	16.40
Amount of contaminant	%	0.10	0.20
Grade of crushed stone	MPa	100	-
Humidity ratio	%		2.20
Fineness modulus	-		1.24

**Table 3 materials-14-04811-t003:** Chemical admixtures used in the experimental investigation.

Type of Chemical Admixture	Description of the Chemical Origin of Admixture
C-3	Synthetic product based on sulphonated naphthalene formaldehyde resin
Dofen	Oligomeric compound based on sodium salt and naphthalene sulfonic acid
ACF-3M	Acetone–formaldehyde resin

**Table 4 materials-14-04811-t004:** The content of chemical admixtures in concrete mixes and (W/C)_)in_ ratio under the same consistence of the concrete mix.

No.	Admixture Type	Content of Admixture (% of Cement Mass)	(W/C)_in_ Ratio
1	C-3	1.00	0.29
2	Dofen	1.00	0.30
3	ACF-3M	0.15	0.37
4	Admixture-free	-	0.37

**Table 5 materials-14-04811-t005:** The detailed plan of the experimental study.

Goals of Experimental Testing	Number of CWD Cycles	Number of Specimens				
		Main Specimens (Including the Admixtures)				Admixture-Free Control Specimens
		C-3	Dofen	ACF-3M	Admixture-Free	
(1) The distribution of specimens by cyclic variations in temperature	75	3	3	3	3	2 × 2 × 4 = 16
	120	3	3	3	3	
(2) The distribution of specimens by integrated action of saline and cyclic variations in temperature	75	3	3	3	3	
	120	3	3	3	3	
Total:		12	12	12	12	16

**Table 6 materials-14-04811-t006:** The chemical composition of 1 m^3^ of saline.

Salts		Salt Content (kg)
Sodium chloride	Na Cl	239.75
Calcium chloride	CaCl × 2H_2_O	49.26
Magnesium chloride	MgCl_2_ × 6H_2_O	34.98
Magnesium sulphate	MgSO_4_ × 7H_2_O	2.89
Potassium bicarbonate	KHCO_3_	0.24
Ammonium chloride	NH_4_Cl	0.24
Total salt content		327.3
Water content		672.2

**Table 7 materials-14-04811-t007:** The main parameters of the cross-sections and the results of short-term axial compression tests on control spun concrete prismatic specimens stored under natural ambient conditions.

Number of CWD Cycles	Type of Chemical Admixture	Code of Prism Specimen	Specimen Wall Thickness *t* (cm)	Compressive Strength *f_c_* (MPa)	Average Compressive Strength ${\bar{f}}_{c}\;(\mathrm{MPa})$	Initial Young’s Modulus *E_c_* (GPa)	Average Initial Young’s Modulus ${\bar{E}}_{c}\;(\mathrm{GPa})$
75	Admixture-free	1141	7.40	55.1	57.2	40.8	39.8
		1146	7.20	59.2		38.8	
	ACF-3M	1415	9.80	60.0	58.1	37.3	36.7
		1417	9.50	56.2		36.0	
	Dofen	1054	9.20	71.7	70.2	41.2	40.0
		1058	9.55	68.7		38.8	
	C-3	1225	9.80	69.0	71.3	38.6	39.5
		1226	9.70	73.6		40.4	
120	Admixture-free	1125	8.25	62.1	59.8	37.4	36.1
		1128	8.05	57.5		34.8	
	ACF-3M	1385	9.55	58.4	60.2	36.5	38.2
		1388	9.50	62.0		39.9	
	Dofen	1065	9.40	69.2	67.8	38.8	38.0
		1068	9.50	66.4		37.2	
	C-3	1235	9.60	70.0	71.8	39.6	38.7
		1238	9.65	73.6		37.8	

**Table 8 materials-14-04811-t008:** Main parameters of the cross-sections and the results of short-term axial compression tests on spun concrete prismatic specimens soaked in water and dried in the air at 45–50 °C following 75 and 120 cycles, respectively.

Number of CWD Cycles	Type of Chemical Admixture	Code of Prism Specimen	Specimen Wall Thickness *t* (cm)	Compressive Strength *f_c_* (MPa)	Average Compressive Strength ${\bar{f}}_{c}\;(\mathrm{MPa})$	Initial Young’s Modulus *E_c_* (GPa)	Average Initial Young’s Modulus ${\bar{E}}_{c}\;(\mathrm{GPa})$
75	Admixture-free	1142	7.45	48.8	48.8	28.86	32.0
		1144	7.50	53.6		-	
		1143	7.40	44.0		35.10	
	ACF-3M	1418	9.90	48.9	52.0	32.30	33.3
		1413	8.95	56.2		33.27	
		1412	9.65	50.8		34.18	
	Dofen	1052	9.75	49.1	58.4	36.58	39,0
		1057	9.70	60.7		41.40	
		1055	9.25	65.3		39.02	
	C-3	1222	9.50	63.4	59.9	39.04	39.1
		1221	9.35	61.4		38.56	
		1224	9.95	55.0		39.65	
120	Admixture-free	1122	8.20	37.0	38.3	18.5	22.0
		1124	8.30	41.7		26.3	
		1126	8.00	36.2		21.2	
	ACF-3M	1382	9.50	40.5	41.5	24.4	26.0
		1384	9.65	39.8		26.8	
		1386	9.60	44.2		26.8	
	Dofen	1062	9.55	54.8	50.8	33.2	29.6
		1064	9.55	50.6		27.6	
		1066	9.60	47.0		28.0	
	C-3	1232	9.65	53.0	56.0	29.4	31.3
		1234	9.70	59.8		34.2	
		1236	9.60	55.2		30.3	

**Table 9 materials-14-04811-t009:** Main parameters of the cross-sections and the results of short-term axial compression tests on spun concrete prismatic specimens soaked in saline and dried in the air at 45–50 °C following 75 and 120 cycles respectively.

Number of CWD Cycles	Type of Chemical Admixture	Code of Prism Specimen	Specimen Wall Thickness *t* (cm)	Compressive Strength *f_c_* (MPa)	Average Compressive Strength ${\bar{f}}_{c}\;(\mathrm{MPa})$	Initial Young’s Modulus *E_c_* (GPa)	Average Initial Young’s Modulus ${\bar{E}}_{c}\;(\mathrm{GPa})$
75	Admixture-free	1148	7.50	62.7	62.3	43.0	42.8
		1145	7.25	64.5		43.8	
		1147	7.15	59.7		41.6	
	ACF-3M	1411	9.50	55.0	53.7	39.8	38.8
		1416	9.35	58.8		37.7	
		1414	9.00	47.2		38.8	
	Dofen	1053	9.00	57,9	62.8	42.0	41.6
		1056	8.90	61.4		40.4	
		1051	9.25	69.0		42.3	
	C-3	1228	9.15	69.4	64.5	41.6	43.0
		1223	9.65	64.2		43.0	
		1227	9.15	59.8		44.5	
120	Admixture-free	1121	8.20	33.9	34.7	26.4	27.4
		1123	8.35	38.0		29.0	
		1127	7.90	32.2		26.8	
	ACF-3M	1381	9.60	35.9	36.1	26.6	27.1
		1383	9.50	33.2		26.1	
		1387	9.75	39.2		28.6	
	Dofen	1061	9.45	50.8	47.7	29.4	29.3
		1063	9.55	45.1		26.4	
		1067	9.60	47.2		32.1	
	C-3	1231	9.60	49.9	52.4	28.2	30.2
		1233	9.65	55.1		32.0	
		1237	9.65	52.2		30.4	

**Table 10 materials-14-04811-t010:** The resistance coefficients of spun concrete *α*_1_ and *β*_1_ subject to the number of wetting cycles in water at a temperature of 20–25 °C and drying in the air at 45–50 °C, considering the type of chemical admixture used.

Type of Chemical Admixture	Resistance Coefficient *α*_1_ (by Equation (1))		Resistance Coefficient *β*_1_ (by Equation (2))	
	Number of Wetting Cycles in Water and Drying in the Air at 45–50 °C		Number of Wetting Cycles in Water and Drying in the Air at 45–50 °C	
	75	120	75	120
Admixture-free	0.85	0.64	0.80	0.61
ACF-3M	0.90	0.69	0.91	0.68
Dofen	0.83	0.75	0.98	0.78
C-3	0.84	0.78	0.99	0.81

**Table 11 materials-14-04811-t011:** The experimental values of the resistance coefficients of spun concrete *α*_2_ and *β*_2_ subject to the number of the cycles of alternate wetting in saline and drying in the air and the temperature gradient considering the type of the chemical admixture used.

Type of Chemical AdMixture	Resistance Coefficient *α*_2_ (by Equation (3))		Resistance Coefficient *β*_2_ (by Equation (4))	
	Number of Wetting Cycles in Saline and Drying in the Air at 45–50 °C		Number of Wetting Cycles in Saline and Drying in the Air at 45–50 °C	
	75	120	75	120
Admixture-free	1.09	0.58	1.08	0.76
ACF-3M	0.92	0.60	1.06	0.71
Dofen	0.89	0.70	1.04	0.77
C-3	0.90	0.73	1.09	0.78

**Table 12 materials-14-04811-t012:** The experimental values of the resistance coefficients of spun concrete to corrosion *α*_3_ and *β*_3_ subject to the number of CWD cycles under the 25 °C temperature gradient of wetting and drying processes.

Type of Chemical Admixture	Resistance Coefficient *α*_3_ (by Equation (5))		Resistance Coefficient *β*_3_ (by Equation (6))	
	Number of CWD Cycles at a Temperature Gradient of 25 °C		Number of CWD Cycles at a Temperature Gradient of 25 °C	
	75	120	75	120
Admixture-free	1.28	0.91	1.34	1.25
ACF-3M	1.03	0.87	1.17	1.04
Dofen	1.08	0.94	1.07	0.99
C-3	1.08	0.94	1.10	0.96

**Table 13 materials-14-04811-t013:** The experimental values of the resistance coefficients of spun concrete to corrosion *α*_3_ and *β*_3_ subject to the number or CWD cycles under the 80 °C temperature gradient of wetting and drying processes.

Type of Chemical Admixture	Resistance Coefficient *α*_3_ by Equation (5)			Resistance Coefficient *β*_3_ by Equation (6)		
	Number of CWD Cycles at a Temperature Gradient of 80 °C			Number of CWD Cycles at a Temperature Gradient of 80 °C		
	25	50	75	25	50	75
Admixture-free	1.00	1.10	0.96	1.03	1.13	1.04
ACF-3M	1.07	0.96	0.91	1.04	0.99	0.93
Dofen	1.00	0.97	0.92	1.24	1.14	1.00
C-3	1.01	0.97	0.92	1.04	1.10	1.03

## Data Availability

Not applicable.

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
