# Peer review of "The Impact of Long-Term Physical Salt Attack and Multicycle Temperature Gradient on the Mechanical Properties of Spun Concrete"

_materials, 2021, doi:10.3390/ma14174811_

Round 1
Reviewer 1 Report
The chemical admixtures, aggressive salt-saturated ground water, absolute temperature, temperature gradient and number of cycles have a significantly impact on the mechanical properties of spun concrete. This manuscript attempts to clarify the long-term influence of these factors on concrete strength. Three types of concrete resistance coefficients are used to evaluate the reduction of elastic modulus and compressive strength of concrete.
This manuscript pointed that the long-term multi cycle effects of temperature gradient and physical salt erosion on concrete compressive strength are negative. This is a well-written paper and the findings are of considerable interest. For the benefit of the reader, there are some points that need to be clarified, and some statements need further demonstration.
1.“…three lateral surfaces were waterproofed with 5 coats of polyvinyl acetate (PVA) glue. Next, the specimens were submerged in specially equipped container filled by saline or water…” The author should add schematic diagram to clarify it.
- The author did not clearly describe the influence of temperature gradient and number of CWD. For example, in Figure.3, when the number of CWD = 75, the jump of data points is obvious, and the author's explanation is vague.
- The author has repeatedly mentioned that microcracks occur on the structural surface (e.g. “… a negative effect of the latter on the mechanical properties of concrete is observed, which is manifested by the formation of micro cracks along the entire structure…”). Generally, the distribution and strike of cracks can prove its damage evolution. Please add a schematic diagram and make a reasonable explanation.
- The steel bars are usually assembled in spun concrete, under combined aggressive ambient conditions (saline groundwater and temperature variations during the day), the corrosion and damage of the spun concrete supporting poles of overhead power transmission lines is serious, the reviewers questioned how the author applied the conclusions of this paper to practice.
- This manuscript proved that improving chemical admixtures has a significant positive impact on the mechanical properties and spun concrete. Please briefly introduce the functions and scope of application of different chemical admixtures before the laboratory tests.
- The necessity of the background in this study can be strengthened. The related works on the corrosion effect of concrete structures should be added to support the statements. The following studies, but not limited, you may refer to improve the introduction.
--Probabilistic life prediction for reinforced concrete structures subjected to seasonal corrosion-fatigue damage. Journal of Structural Engineering, 2020, 146(7): 04020117;
--Predicting corrosion fatigue crack propagation behavior of HRB400 steel bars in simulated corrosive environments. Journal of Materials in Civil Engineering, 2021, 33(6): 04021127.
Author Response
Dear Reviewer,
Thank you for your valuable comments that have contributed to improving the quality of the article. Article has been revised and a number of amendments and corrections have been made.
The corrections were made as follow:
1) The description of coating of cutted surfaces is a little extended in text (275 line). In our opinion, the description in the text is sufficienty clear and, probably, the schematic diagram is not necessary.
2) The explanation of the “jump” in the Fig. 1 is presented in text (lines 394-414).
3) In our opinion, this problem is explained in text (514-539) and in the conclusions (578-587). The extended extra text is inserted for better clarity. (469-474)
4) Based on the results of investigations presented the recommendations for RC factories and designers were provided to use chemical admixtures, that reduce the w/c ratio of the concrete mix and increase the density and durability of concrete (for broader info please see Kliukas et al [36]).
5) The chemical admixtures analyzed (especially C-3 and Dofen), have been widely used for vibrated concrete mixtures. While our investigations are related with spun concrete namely. And such studies are lacking (for broader info please see Kliukas et al [36]).
6) This suggestion is partially accepted.
Best Regards,

Reviewer 2 Report
The authors investigated the spun concrete made with different chemical admixtures under long-term exposure to aggressive salt-saturated ground water and a cyclic temperature gradient. Specimens were made and tested under multi-cycle processing under combined aggressive ambient conditions. It is found in the research that Young's modulus and durability of concrete were adversely affected and chemical admixtures can improve the mechanical properties and durability of spun concrete. It is an interesting work. But the current presentation is not comprehensive enough. I have several comments before I can suggest publishing of this work.
-it is mentioned that chemical admixtures improve the structure of spun concrete. But the authors need to express the detailed reasons and principles of this conclusion.
- How to determine the number of cycles and the interval of temperature, and whether it has an impact on the results.
-the establishment of resistance of spun concrete to corrosion should be detailed.
-Several recent publications are suggested for example, DOI: 10.1016/j.conbuildmat.2020.118048
Author Response
Dear Reviewer,
Thank you for your valuable comments that have contributed to improving the quality of the article. Article has been revised and a number of amendments and corrections have been made.
The corrections were made as follow:
1) Partially, this problem is explained in text (514-523) and in the conclusions (593-596). While more extended analysis of the effectiveness of chemical admixtures on to spun concrete (including microstructure, W/C ratio…..) is presented in previous investigations of the authors (appropriate references given in text). Please see Kliukas et al [36].
2) In the real operating conditions of structures, it is very difficult to determine the exact number of CWD and temperature differences, at which the noticeable destruction of the concrete of the structure begins. Probabilistic methods should be used for these purposes. The carried out research to be continued in that direction.
3) In our opinion it is described in chapter No 3 “The assessment of the mechanical properties of the specimens” (278-362).
4) The References list is partially extended, according to suggestions of reviewers.
Best Regards,

Reviewer 3 Report
Dear authors,
Thank you very much for this highly practical and beneficial scientific intention. A topic is needed and it is important to do these studies.
The first three paragraphs lack references. Even if it is true factual information, you have to assign real literature to it.
The rest of the introduction is written in order and corresponds to the content of the article.
I suggest adding more literature right here:
10.1186 / s40069-018-0235-x
10.3311 / PPci.13780
10.14359 / 51716678
The description of the basic material is OK.
I have a question about water:
You write "Potable water from water supply system was used for producing the concrete mix. Water quality plays a crucial role in the strength of concrete." - Any quote? reason? another explanation?
Why did you want concrete as without any admixture and not straight concrete that really doesn't have those admixtures? What does this mean if you evaluate real column constructions?
The description of the production is adequate.
The description of the results and the discussions are extensive and I appreciate that they are detailed and well done.
The conclusions are clear, but they are not exposed to any other available literature - do they correspond to other studies?
Regards,
Author Response
Dear Reviewer,
Thank you for your valuable comments that have contributed to improving the quality of the article. Article has been revised and a number of amendments and corrections have been made.
The corrections were made as follow:
1) On the 1st and 2nd paragraphs the general knowledges according to corrosion of concrete are presented. Beacause it is repeatedly reviewed and analysed in detail later, the references are indicated namely at the place of more detailed analysis. The reference is mentioned in paragraph 3.
2) We apologize for the ambiguity. In our country, the water from water supply system is of very high quality, free of salts and other mineral impurities, which is rare in other countries. The line 172 is removed.
3) The References list is partially extended, according to suggestions of reviewers.

Round 2
Reviewer 2 Report
My comments are well answered. I suggest publishing.Reviewer 3 Report
Dear authors,
Thank you very much for the repairs made.
Thanks to these fixes and changes, the article has been moved to a higher level.
The knowledge and results you have acquired are suitable for publication.
Regards